# Thermal Environment Design of Outdoor Spaces by Examining Redevelopment Buildings Opposite Central Osaka Station

**Hideki Takebayashi** 

Department of Architecture, Kobe University, Nada Ward 657-8501, Japan; thideki@kobe-u.ac.jp;
Tel.: +81-78-803-6062

**Abstract:** Thermal environmental design in an outdoor space is discussed by focusing on the proper selection and arrangement of buildings, trees, and covering materials via the examination of redevelopment buildings in front of Central Osaka Station, where several heat island countermeasure technologies have been introduced. Surface temperatures on the ground and wall were calculated based on the surface heat budget equation in each 2 m size mesh of the ground and building wall surface. Incident solar radiation was calculated using ArcGIS and building shape data. Mean radiant temperature (MRT) of the human body was calculated using these results. Distribution of wind velocity was calculated by computational fluid dynamics (CFD) reproducing buildings, obstacles, trees, and the surroundings. The effect of MRT on SET* was greater than that of wind velocity at 13:00 and 17:00 on a typical summer day. SET* reduction was the highest by solar radiation shading, followed by surface material change and ventilation. The largest ratio of the area considered for the thermal environment was 83% on Green Garden, which consists of 44% of building shade, 21% of tree shade, 7% of water surface, and 11% of green cover. It is appropriate to consider the thermal environment design of outdoor space in the order of shade by buildings, shading by trees, and improvement of surface materials.

**Keywords:** outdoor space; thermal environment; radiation environment; wind environment

## 1. Introduction

In our previous study [1], the effects of solar radiation shading by trees in open spaces was evaluated through a case study. Outdoor open spaces are used for various purposes such as walking, resting, talking, meeting, studying, exercising, playing, performing, eating, and drinking. Therefore, providing various thermal environments according to the various purposes above-mentioned is desirable. The results from one of our previous studies [2] are reprinted as follows: "By investigating the redevelopment building in front of Central Station in Osaka, the radiation environment was evaluated with a focus on ground cover materials and solar radiation shielding. ArcGIS and building shape data were used to calculate the spatial distribution of solar shading. A surface heat balance equation was calculated to determine the surface temperature of the ground and walls. Assuming the human body is a sphere, the mean radiation temperature (MRT) of the human body was calculated. Solar radiation shielding and improvements in surface coverage were the most dominant factors in the radiation environment. On a typical summer day (August) when air temperature is high, improvements in solar shading and surface coverage did not provide a comfortable standard new effective temperature (SET*) in the afternoon. However, there were several places where people did not feel uncomfortable, especially in the rooftop garden and green gardens, which have large areas of shaded grass and water."

This study is the continuation of our previous studies [1,2]. The study site and the calculation method of solar radiation, surface temperature, MRT, and wind velocity distribution were the same as those of our previous studies [1,2]. Results from another of our previous studies [1] are reprinted as follows. "At 10:00, 13:00, and 17:00 on a typical summer sunny day, we analyzed building and tree awnings at 25 and 32 measurement points in Station Plaza and Green Garden. Assuming various heights of buildings, the need for sunshade by trees was pointed out at 10 m or more from the south building and 6 m or more from the west or east building." The subject of this study was the thermal environmental design in an outdoor space, focusing on the proper selection and arrangement of buildings, trees, and covering materials through the examination of redevelopment buildings in front of Central Osaka Station where several heat island countermeasure technologies such as terrace gardens on medium height rooftops (Rooftop Gardens), mist and waterscape in Station Plaza, ground garden with trees, water, and green cover between buildings (Green Garden), rows of trees, and water streams around the buildings were introduced. In particular, the results of the case study were analyzed from the perspective of how effective it is to proceed with the thermal environmentally-friendly design of outdoor spaces to increase the generic understanding for better outdoor thermal environment design.

## 2. Calculation of Thermal Element Distribution

The study site layout is shown in Figure 1 [3], which is the same as those in our previous studies [1,2]. The layouts of Station Plaza, Rooftop Gardens, and Green Garden are shown in Figure 2. The ratio of each ground cover type is shown in Table 1. An outline of the study sites is shown in Table 2. The trees were reproduced based on the actual situation, and the average height of the trees was about 6 m because this study was conducted just after completion of the site. The calculation methods for surface temperature, solar radiation, MRT, and wind velocity distribution were also the same as those used previously [1,2], and an outline of the calculation methods is shown in Table 3. Daytime air temperatures obtained from the Osaka Meteorological Observatory on a typical summer day (11 August 2013), the day of the autumnal equinox (23 September 2013), and the day of the summer solstice (21 June 2013) are shown in Figure 3. The air temperature was over 30 °C in the morning and over 35 °C in the afternoon on a typical summer day, and it was around 30 °C in the afternoon on the day of the autumnal equinox and day of the summer solstice. MRT was calculated by integrating the amount of solar radiation and infrared radiation incident on the human body. The incident solar radiation was calculated by the method described above, and the incident infrared radiation was calculated using the surface temperature and the view factor of the surrounding objects. The objective area was divided into meshes according to the form of the buildings. The calculation conditions for computational fluid dynamics (CFD) are shown in Table 4, referring to Tominaga et al. [4]. The applicability of this software for an urban area such as Osaka City was verified using a verification database provided by Tominaga et al. [4].

**Table 1.** Ratio of each ground cover type.

| Site | Concrete | Wood Deck | Grass | Water Surface | Asphalt |
|---|---|---|---|---|---|
| Station Plaza | 79% | 0% | 0% | 10% | 11% |
| Rooftop Gardens | 37% | 25% | 38% | 0% | 0% |
| Green Garden | 40% | 0% | 26% | 15% | 19% |
| Total | 52% | 5% | 12% | 6% | 25% |

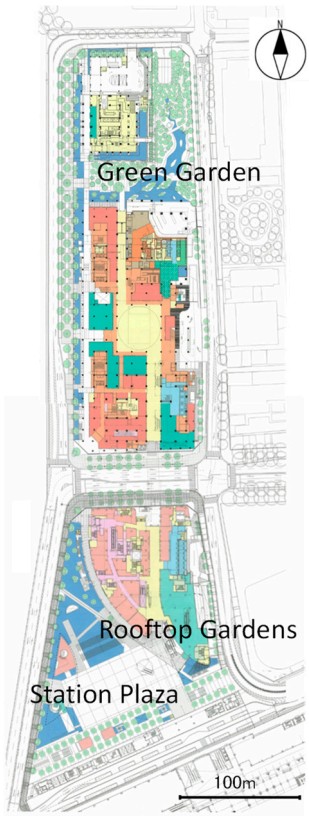

**Figure 1.** Layout of the study site [1,2].

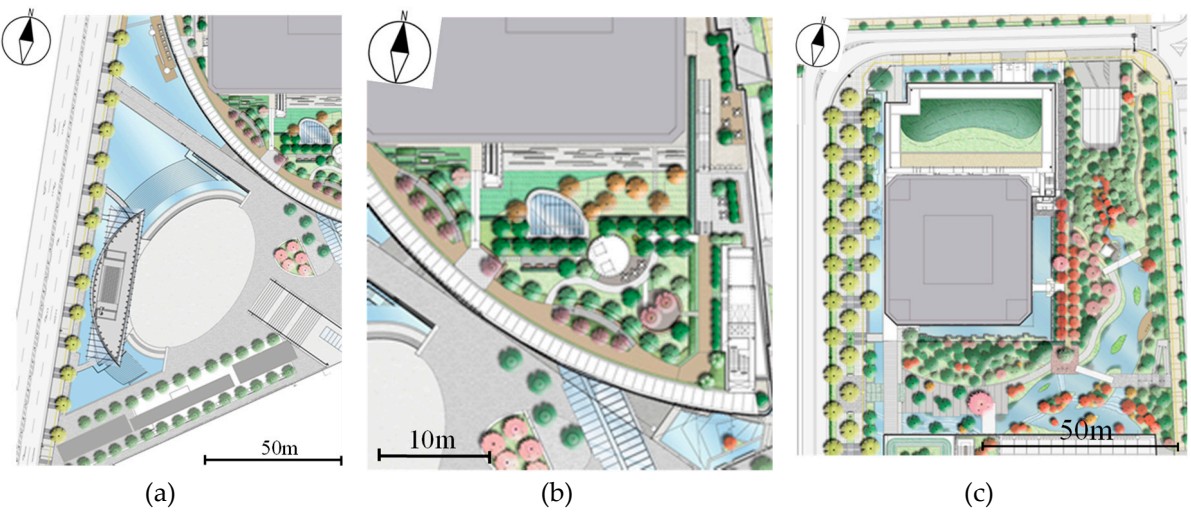

|         (a)         |         (b)         |         (c)         |

**Figure 2.** Layout of (**a**) Station Plaza; (**b**) Rooftop Gardens; and (**c**) Green Garden [1,2].

**Table 2.** Outline of the study sites.

| Study Sites | Location | Land Cover Characteristics |
|---|---|---|
| Station Plaza | The site is beside the north-eastern (180 m high) and the southern (150 m high) high-rise buildings. | There is little vegetation cover, and open spaces (concrete surfaces) and water surfaces dominate. |
| Rooftop Gardens | The site is on the southern (45 m high) and the central (43 m high) middle-rise buildings. | The ratios of concrete, wood deck, and grass are similar, ~30%. |
| Green Garden | The site is between the northern (174 m high) and the central (154 m high) high-rise buildings. | The site features green grassy areas, water surfaces, medium-height trees, and concrete walkways. |

**Table 3.** Outline of calculation methods, which is a reprint of our previous study [1,2].

| Element | Method |
|---|---|
| Surface temperature | It is calculated based on the surface heat budget equation in each 2 m size mesh of the ground and building wall. Air temperature, air absolute humidity, underground temperature, convection heat, and moisture transfer coefficients of the function of wind velocity are set by the observation values as boundary conditions. |
| Incident solar radiation | It is calculated using ArcGIS and building shape data, as per the method described by Takebayashi et al. [5]. The visible area of the upper hemisphere is calculated by ArcGIS tool considering the influence of the adjacent buildings. The visible area is then overlain with the sun-map and sky-map raster to calculate the diffuse and direct solar radiation received from each direction. |
| Mean Radiant Temperature | MRT of the human body is calculated using surface temperature and incident solar radiation. The human body is assumed to be a sphere, and solar radiation absorption ratio of the human body is assumed to be 0.5, considering the clothing conditions in summer. |
| Wind velocity | It is calculated by computational fluid dynamics (CFD) reproducing buildings, obstacles, trees, and the surroundings. The standard k-ε turbulence model (one of the Reynolds–Averaged Navier–Stokes equation (RANS) models) is selected for use in the simulation. A general purpose CFD software (STREAM, version 9, Software Cradle Co. Ltd., Osaka, Japan) is used for calculation. The Navier–Stokes equations are discretized using a finite volume method, and the SIMPLE algorithm is used to handle pressure-velocity coupling. Inflow boundary conditions are given based on weather conditions. |

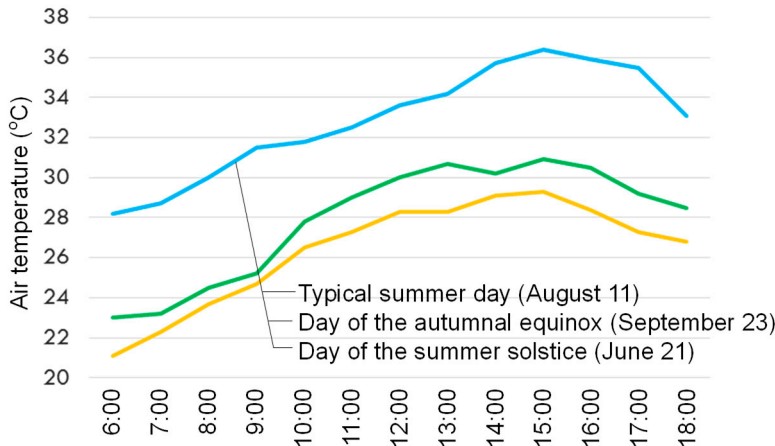

**Figure 3.** Daytime air temperature at Osaka Meteorological Observatory on a typical summer day (11 August 2013), the day of the autumnal equinox (23 September 2013), and the day of the summer solstice (21 June 2013).

**Table 4.** The calculation conditions for computational fluid dynamics (CFD).

| Software | STREAM ver. 9 |
|---|---|
| Turbulence model | Standard k-ε model |
| Advection term | Up-wind difference scheme |
| Inflow boundary | Power low, 3.9 m/s, WSW at 50.9 m high, power: 0.27 |
| Outflow boundary | Zero gradient condition |
| Up, side boundary | Free-slip condition |
| Wall, ground surface | Generalized log-low |
| Convergence criterion | $10^{-5}$ |

The calculation results of surface temperature, MRT, wind velocity, and SET* distribution at 13:00 on a typical summer day (11 August 2013) are shown in Figure 4. Surface temperature and MRT

distributions are reprints of our previous study [2]. The explanation concerning SET* is the same as per a previous study [2] and is reprinted as follows: "The equivalent dry bulb temperature in an isothermal environment with a relative humidity of 50% is the definition of SET* [6]. The subject has the same thermal stress and temperature regulation strain as the actual test environment while wearing standardized clothing for the relevant activity. It is used for thermal environmental evaluation. Gagge et al. proposed SET* by improving the new effective temperature (ET*) [6]. Gagge et al. also proposed ET*, an index based on human energy balance and a two-node model [7]. SET* is frequently used as an indoor and outdoor comfort indicator. The metabolic rate was assumed to be 2.0 met and the clothes was assumed to be 0.6 clo. The thermal equilibrium calculation program for the thermos-physiological model of the human body, which has already been verified in previous studies [6,7], calculates SET*." Air temperature and relative humidity were set by the observation values at the Osaka Observatory. We compared the temporal and spatial distributions of MRT in our previous study [2] from 5:00 to 18:00 on the day of the summer solstice, a typical summer day in August, and the day of the autumnal equinox. The result is reprinted as follows; "Incident solar radiation dominates the characteristics of MRT spatial distribution. Sunny and shaded points cause large differences in MRT time changes. In the afternoon of the summer solstice and autumnal equinox, a comfortable thermal environment was realized by sun shade. However, on a typical summer day (August), since air temperature is too high, it is difficult to make SET* comfortable in the afternoon, both with sun shade and with improved surface cover." In this study, 13:00 on a typical summer day (11 August 2013) was chosen as a representative time, together with 17:00 on a typical summer day (11 August 2013) and 13:00 on a summer solstice day (21 June 2013). Weak wind regions affected by buildings, obstacles, and planting were confirmed in the wind velocity distribution. The influence of MRT was dominant in the SET* distribution, despite the high wind velocity in Rooftop Gardens, which is shown in the upper right in Figure 4c as GL + 46.5 m.

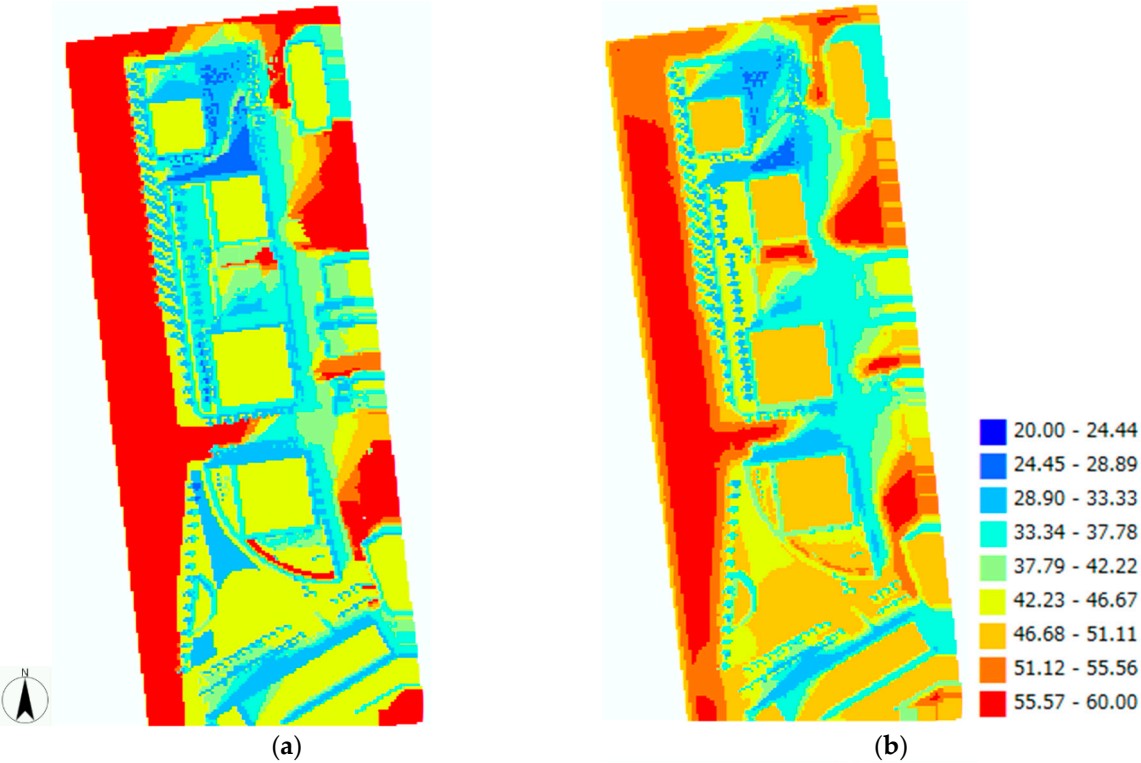

(**a**)　　　　　　　　　　　　　　　　　　　　　　　　(**b**)

**Figure 4.** *Cont.*

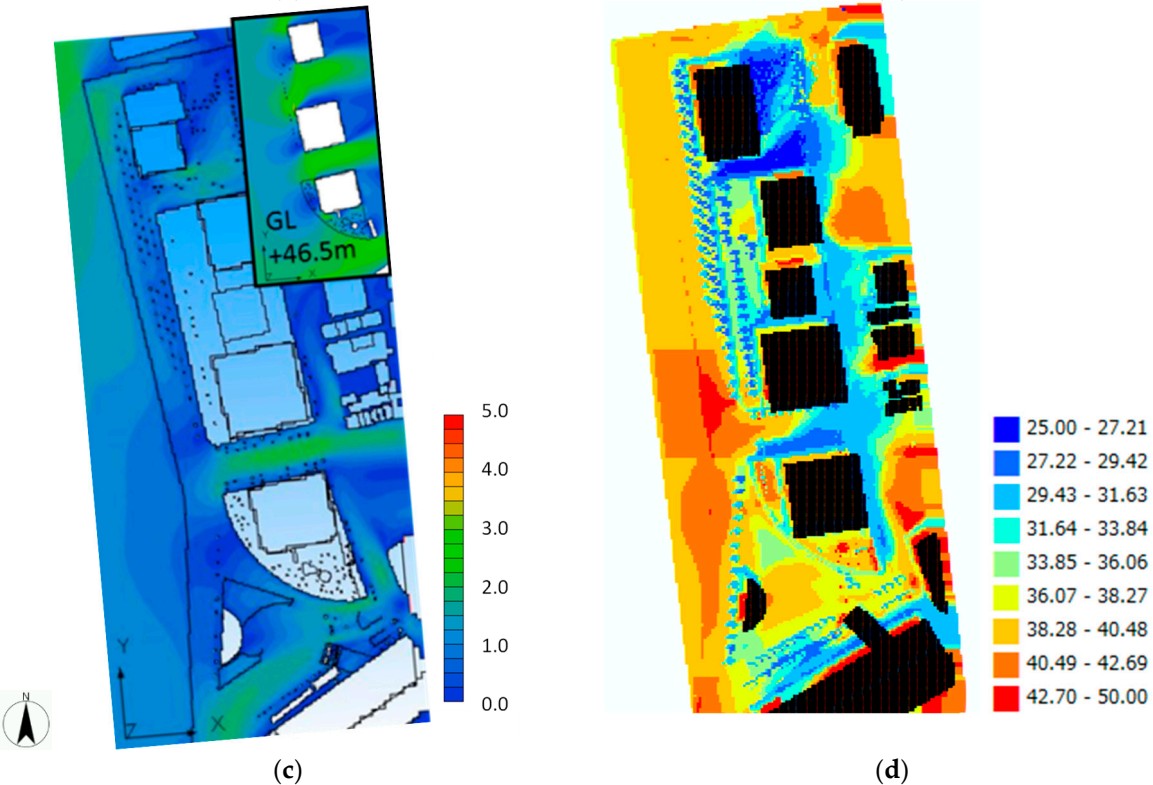

**Figure 4.** Calculation results of (**a**) surface temperature (°C), (**b**) mean radiant temperature (MRT) (°C), (**c**) wind velocity (m/s), and (**d**) standard new effective temperature (SET*) (°C) distribution at 13:00 on the typical summer day (11 August 2013).

## 3. Relationship between MRT, Wind Velocity, and SET*

### 3.1. Evaluation under Various Conditions

The relationships between MRT, wind velocity, and SET* at 13:00 and 17:00 on a typical summer day (11 August 2013) and at 13:00 on the summer solstice day (21 June 2013) are shown in Figure 5. Air temperature is high (34.0 °C) at 13:00 and is still high (33.6 °C) at 17:00 on the typical summer day. Furthermore, it was a little low (28.3 °C) at 13:00 on the summer solstice day. The average wind velocity and MRT values in sunny and shaded locations on Station Plaza, Rooftop Gardens, and Green Garden are presented by the vertical and horizontal axes. The standard deviation is expressed by the length of the bar, and the number of corresponding points is expressed by the size of bubbles. The numbers inside and beside the bubbles indicate the number of corresponding points. SET*, which is the center of the bubble, is recognized from the background contour lines. Air temperature and relative humidity given uniformly, are also shown in the figure. The translucent bubbles denote sunny points, and opaque bubbles denote shaded points. The relationship between SET* and thermal comfort evaluation reported by Ishii et al. [8] is as follows: comfortable < 26.5 °C < slightly comfortable < 27.5 °C < neither comfortable nor uncomfortable < 29.5 °C < slightly uncomfortable < 31.5 °C < uncomfortable < 32.5 °C < very uncomfortable. The comfortable range is shown in the blue colored background with 29.5 °C as the boundary in Figure 5, which is the boundary between neither comfortable nor uncomfortable and slightly uncomfortable by Ishii et al. [8].

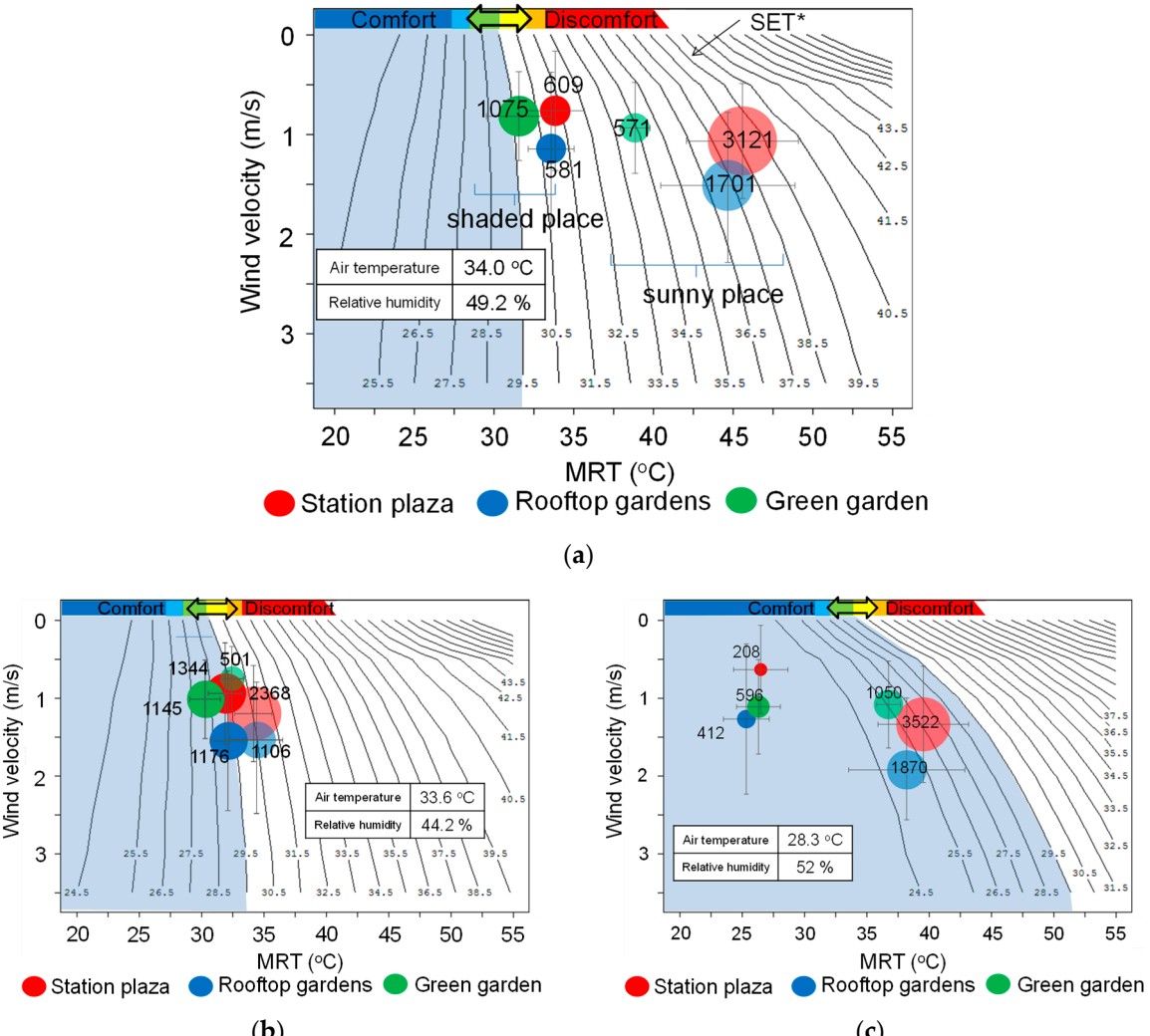

**Figure 5.** Relationship between MRT, wind velocity, and SET*. (**a**) 13:00 on the typical summer day (11 August 2013); (**b**) 17:00 on the typical summer day (11 August 2013); (**c**) 13:00 on the summer solstice day (21 June 2013).

The effect of MRT on SET* was greater than that of the wind velocity at 13:00 and 17:00 on the typical summer day. SET* values on both sunny and shaded points are in a relatively comfortable range at 13:00 on the summer solstice day because of the relatively lower air temperature. The difference in MRT values between the sunny and shaded points was about 8 to 12 °C, so the difference in SET* values was also large, which was about 5 to 8 °C at 13:00 on the typical summer day. While the ratio of shaded points was small at Station Plaza and Rooftop Gardens, it was slightly higher in Green Garden. As a result, the number of points with low surface temperature was slightly high, so the averaged MRT and SET* were low, even in sunny points at Green Garden. The difference in wind velocity between Rooftop Gardens, Station Plaza, and Green Garden was approximately less than 1.0 m/s, so the difference in SET* values was slightly lower at sunny points. SET* approached a comfortable range at all sites, especially in shaded points, at 17:00 even on the typical summer day. Although the difference in SET* values between sunny and shaded points was large, the SET* values at any site were in a comfortable range at 13:00 on the summer solstice day.

### 3.2. Influence of Surface Materials

The relationship between MRT, wind velocity, and SET* on Station Plaza, Rooftop Gardens, and Green Garden at 13:00 on the typical summer day (11 August 2013) is shown in Figure 6. While the difference in SET* values between sunny and shaded places was about 8 °C at Station Plaza, it was only 1 to 2 °C between the concrete and water surfaces. The SET* values on sunny wooden decks in Rooftop Gardens were high because of the high MRT. The differences between sunny and shaded places on wooden decks, concrete, and green cover were 6 to 9 °C. While SET* on the shaded green cover was a little lower than that on shaded concrete, it was almost the same on the sunny green cover and sunny concrete because the difference in MRT was small due to their mixed presence on the slightly narrow Rooftop Gardens. While the difference in SET* values between sunny and shaded places was about 4.5 to 6 °C in Green Garden, it was only 1 to 2.5 °C between the water surface, green cover, and concrete. Summaries of SET* reduction by solar radiation shading, surface material change, and ventilation are shown in Table 5. SET* reduction was the highest by solar radiation shading, followed by surface material change and ventilation.

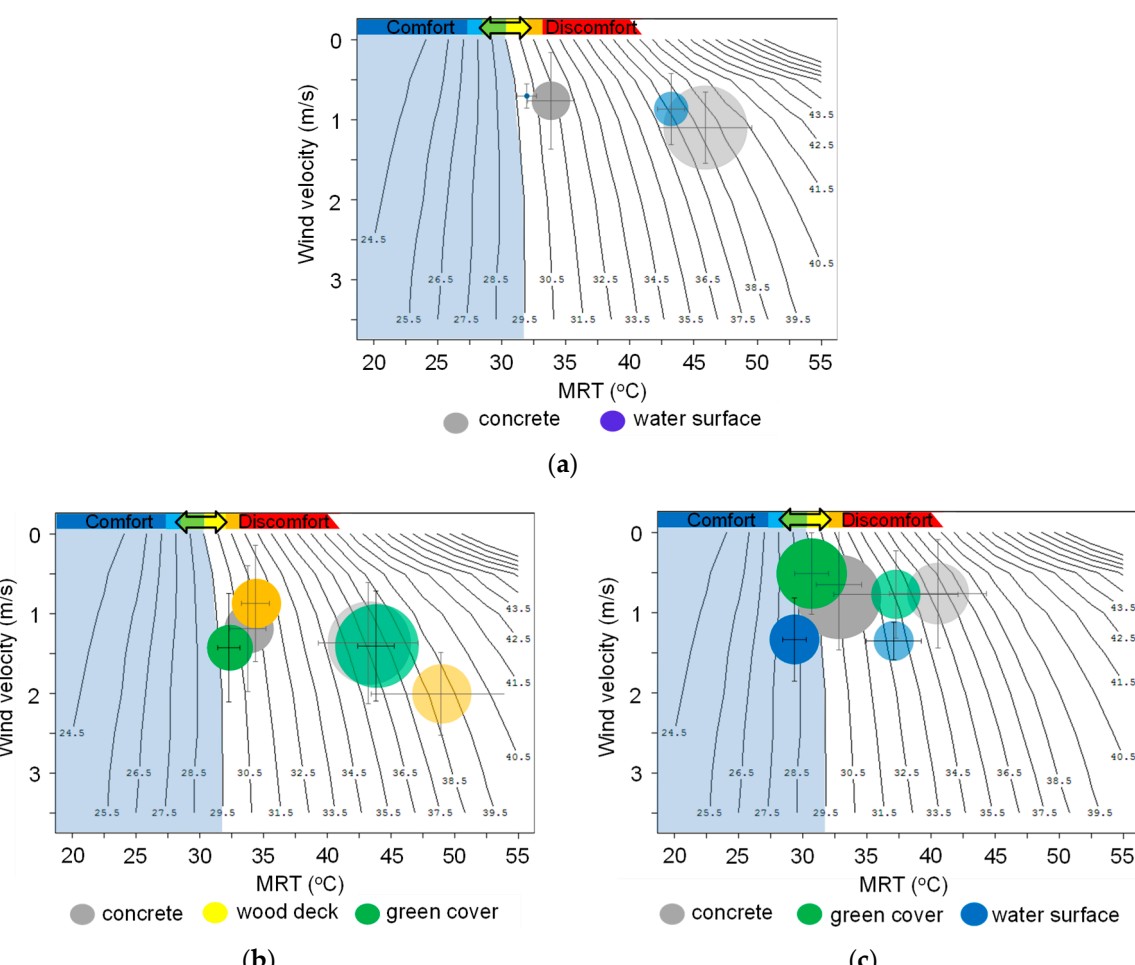

**Figure 6.** Relationship between MRT, wind velocity, and SET* at 13:00 on the typical summer day (11 August 2013). (**a**) Station Plaza; (**b**) Rooftop Gardens; and (**c**) Green Garden.

**Table 5.** Summaries of SET* reduction by solar radiation shading, surface material change, and ventilation.

| SET* Reduction | Station Plaza | Rooftop Gardens | Green Garden |
|---|---|---|---|
| Solar radiation shading | 8.0 °C | 6.0 °C | 6.0 °C |
| Surface materials change from concrete | to water surface 2.0 °C (sunny) 1.0 °C (shade) | to green cover 0 °C (sunny) 1.5 °C (shade) | to green cover 2.5 °C (sunny) 1.0 °C (shade) to water surface 2.5 °C (sunny) 1.5 °C (shade) |
| Ventilation | 0.3 °C | 0.5–1.5 °C | 0 °C |

## 4. Discussion

Changes in shaded areas by buildings, trees, and surface cover to water surface and green cover on Station Plaza, Rooftop Gardens, and Green Garden are shown in Figure 7. The outer circle indicates the shaded area by buildings in blue, the middle circle indicates the shaded area by trees in green, and the inner circle indicates water surface in light blue and green cover in light green. Finally, areas where these were not considered (sunny places) are indicated in red in the inner circle.

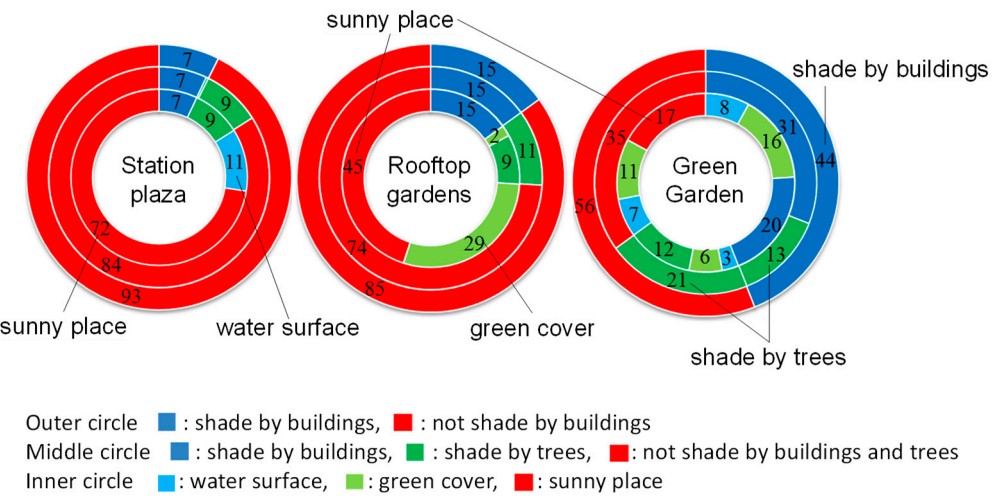

**Figure 7.** Shaded area by buildings and trees and surface cover change to water surface and green cover on Station Plaza, Rooftop Gardens, and Green Garden.

Shaded areas by trees are required on Station Plaza and Rooftop Gardens because shaded areas by buildings (7% and 15%, respectively) were much smaller than those in Green Garden (44%). As a result, as shown in Figure 4, SET* values at Green Garden were a little mitigated than those in other sites. This is in accordance with the results of previous studies by Ali-Toudert and Mayer [9,10], who showed that shading is the key strategy to mitigating outdoor heat stress under hot summer conditions. However, the shading effect by trees does not completely contribute to the shaded area at Green Garden because shaded areas by trees are located behind buildings (13%) and are included in shaded areas by buildings (44%). As a result, shaded areas by trees were 34% in all, but trees contributed to only 21% appearance of shade in Green Garden. This is consistent with the results of Algeciras et al. [11], who showed that the spatial distribution of thermal conditions at the street level depends strongly on the aspect ratio and street direction. Nevertheless, shaded areas by trees on Station Plaza and Rooftop Gardens (9%, 11%) were smaller than those at Green Garden (21%). Tree growth and tree arrangement have important effects on the thermal environment of the outdoor space, as pointed out by Liang et al. [12]

and Zhao et al. [13], respectively. Although previous researchers such as Ali-Toudert and Mayer [10], Lee et al. [14], and Chen et al. [15] have pointed out that building geometry and vegetation play the most significant role in affecting the thermal comfort index, unfortunately there is an insufficient number of trees at Station Plaza and Rooftop Gardens. Therefore, surface material changes such as water surface and green cover is also required at Station Plaza and Rooftop Gardens. Finally, water surface (11%) was 27% of the area considered for the thermal environment at Station Plaza (7% of building shade, 9% of tree shade, and 11% of water surface), and green cover (29%) was 55% of the area considered for the thermal environment at Rooftop Gardens (15% of building shade, 11% of tree shade, and 29% of green cover). As a result, as shown in Figure 6 and Table 5, the SET* values at Station Plaza and Rooftop Gardens were mitigated by surface material changes. In other words, the effect of the surface material countermeasures shown in Figure 2 was added in this process. Therefore, it is appropriate to consider the thermal environment design of outdoor spaces in the order of shade by buildings, shade by trees, and improvement by surface materials. The largest ratio of the area considered for the thermal environment was 83% at Green Garden, which consists of 44% of building shade, 21% of tree shade, 7% of water surface, and 11% of green cover.

## 5. Conclusions

Thermal environmental design in outdoor space is discussed by focusing on proper selection and arrangement of buildings, trees, and covering materials via the examination of redevelopment buildings in front of Central Osaka Station, where several heat island countermeasure technologies have been introduced. The effect of MRT on SET* was greater than that of wind velocity at 13:00 and 17:00 on a typical summer day. SET* values on both sunny and shaded points were in a relatively comfortable range at 13:00 on the summer solstice day because of the relatively lower air temperature. SET* reduction was the highest by solar radiation shading (about 6 to 8 °C at 13:00 on the typical summer day), followed by surface material change (about 0 to 2.5 °C at 13:00 on the typical summer day) and ventilation (about 0 to 1.5 °C at 13:00 on the typical summer day). From the analysis of shaded area by buildings and trees and surface cover change to water surface and green cover at Station Plaza, Rooftop Gardens, and Green Garden, the largest ratio of the area considered for the thermal environment was 83 % at Green Garden, which consists of 44% of building shade, 21% of tree shade, 7% of water surface, and 11% of green cover. Areas shaded by trees are required at Station Plaza and Rooftop Gardens because the shaded area by buildings (7% and 15%, respectively) was much smaller than that at Green Garden (44%). Furthermore, because the shaded area by trees at Station Plaza and Rooftop Gardens (9% and 11%, respectively) was smaller than that at Green Garden (21%), surface material changes such as water surface and green cover are also required at Station Plaza and Rooftop Gardens. It is appropriate to consider the thermal environment design of outdoor space in the order of shade by buildings, shading by trees, and improvement of surface materials.

**Author Contributions:** Conceptualization, H.T.; investigation, H.T.

**Funding:** This research received no external funding.

**Acknowledgments:** The author thanks Makiko Kasahara, Shingo Tanabe, and Makoto Kouyama for their cooperation with respect to our analysis.

**Conflicts of Interest:** The authors declare no conflicts of interest.

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
