# Peer review of "Thermal Environment Design of Outdoor Spaces by Examining Redevelopment Buildings Opposite Central Osaka Station"

_climate, doi:10.3390/cli7120143_

Round 1

Reviewer 1 Report

The article presents the modeling of climatic parameters aiming to indicate architectural projects that reduce the effects of the urban heat island.

Need to provide more details:

1) From previous experiments mentioned in references 1 and 8;

2) Need to characterize procedures for estimating incident global solar radiation. In the text it is indicated as using ARCGIS.

3) Define in more detail what represents the “Standard new Effective Temperature (SET *)”;

4) The resolution of the figures should be improved, in particular figure 3.

5) Necessary to include data observed in weather station installed near the study area. As it stands no weather attribute has been measured effectively.

6) What weather conditions occurred on the typical modeling day?

7) It is necessary to include other references in the item “discussion”.

8) Define which heat island mitigation measures were adopted around the study area.

9) present which negative points of the adoption of green areas as heat island mitigators.

10) In the bibliographic references look for other authors who studied the subject.

11) Expand the modeling period to other typical days in other seasons as well as to other times of the day.

Author Response

Thank you very much for your insightful comments and suggestions. I have made the changes and prepared a pointwise response to all your comments. Please see the attachment.

Reviewer 2 Report

Overall

The topic of thermal environment in design of outdoor spaces is highly relevant keeping in mind major megatrends of climate change and urbanization. We need plenty of case studies, like this one in Central Osaka Station, in order to increase generic understanding on how to apply climate information in design of built environment for the benefit of well-being of people.

This manuscript reports impacts of solar radiation and shade, ground cover type and wind on experienced thermal comfort on typical summer day and summer solstice day in fine spatial resolution using sophisticated methods. The outcomes are interesting and worth publishing in Climate, but the manuscript requires major revision in order to improve readability of it. My concerns are not related to methods or outcomes, which I believe I mostly succeeded to interpret the way the author meant, but the language is clearly a problem.

I suggest proper language proof. The topic is so complex that I believe there may be a need for couple of iteration rounds between the author and native English language expert in order to make sure that the text is easy to read and interpret without scientific mistakes. I think suggesting better sentences and expressions throughout the manuscript would be so time-consuming that it is not a task of a reviewer.

Below some comments for revision - in addition to the language proof

- This study in continuation for couple of previous studies of the author. It would be useful to summarize their main outcomes in more detail – either in the Introduction or in the Supplementary material (which is pretty useless in the current version).

- A short description of the climate in Osaka would help readers to put this study in the wider context when thinking how representative the outcomes would be in their climate.

- Sub-titles under Chapter 3 (results) would improve readability. The part of the chapter starting from line 118 describes how thermal comfort varies in time in the study area. And the part starting from line 149 describes the impact of surface type on the thermal comfort.

- In Figures 4 and 5 the blue colored background obviously represents thermally comfortable circumstances. Would be good to mention in the text – e.g. on line 129.

- Explain also in the text what the numbers inside and beside the bubbles in the figures mean.

- I’m not sure what the messages of Figure 6 in the Discussion are... Consider providing this information already in Chapter 2 describing the study site.

Author Response

(The authors gave the same response as above.)

Reviewer 3 Report

Dear authors,

The topic of this paper is of interest as the outdoor thermal environment design is a critical issue with the problem of urban overheating. However, I have several concerns on this paper. 

1). The title mentions 'focusing on radiation and wind environment'. However, the introduction has only presented the shading effects on the outdoor thermal environment (outdoor thermal comfort). It is essential to mention what are the wind (or ventilation you mentioned) relationship with the urban typology, and further its potential impact on the outdoor thermal environment. 

2) The research gap has not been well presented in the introduction, and what  the original contribution is should be presented.

3) For the study site layout, what is the vertical profile (building height, tree height)?

4) Any validation and calibration of the RANS model? What is the criteria for model convergence? What are the specific input? 

5) Pls present the method to calculate MRT.

6) Figure 5, what are the criteria for the comfort and discomfort? 

Overall, this is an interesting paper. It could be acceptable, if authors can address above questions.

Author Response

(The authors gave the same response as above.)

Round 2

Reviewer 2 Report

In this revised version of the manuscript, my previous comments have been adequately taken into account: the improvements in the language of the manuscript make it more reader-friendly, and the results and conclusions are now more clearly presented.

Reviewer 3 Report

Well done.